# Development of Fatty Acid Reference Ranges and Relationship with Lipid Biomarkers in Middle-Aged Healthy Singaporean Men and Women

**DOI:** 10.3390/nu13020435

**Published:** 2021-01-29

**Authors:** Cody A. C. Lust, Xinyan Bi, Christiani Jeyakumar Henry, David W. L. Ma

**Affiliations:** 1Department of Human Health and Nutritional Sciences, University of Guelph, Guelph, ON N1G 2W1, Canada; clust@uoguelph.ca; 2Clinical Nutrition Research Centre (CNRC), Singapore Institute of Food and Biotechnology Innovation (SIFBI), Agency for Science, Technology and Research (A*STAR), 14 Medical Drive #07-02, MD 6 Building, Singapore 117599, Singapore; bi_xinyan@sifbi.a-star.edu.sg (X.B.); jeya_henry@sifbi.a-star.edu.sg (C.J.H.); 3Department of Biochemistry, Yong Loo Lin School of Medicine, National University of Singapore, Singapore 117599, Singapore

**Keywords:** fatty acids, lipids, omega-3, cholesterol, triglycerides, reference range, Singapore

## Abstract

Dietary fatty acids (FA) are essential for overall human health, yet individual FA reference ranges have yet to be established. Developing individual FA reference ranges can provide context to reported concentrations and whether an individual displays deficient, or excess amounts of FA. Reference ranges of sixty-seven individual FA (μmol/L) were profiled and analyzed using gas chromatography with a flame ionization detector from serum samples collected from 476 middle-aged Singaporean males (BMI:23.3 ± 2.9) and females (BMI:21.8 ± 3.6). Measures of triglycerides (TG), high-density lipoprotein (HDL), low-density lipoprotein (LDL), and total cholesterol (TC) (mmol/L) were also collected. The mean FA concentration seen in this cohort (11,458 ± 2478 was similar to that of overweight North American cohorts assessed in past studies. Ten biologically relevant FA were compared between sexes, with females exhibiting significantly higher concentrations in four FA (*p* < 0.05). A multiple regression model revealed the ten FA contributed significantly to nearly all lipid biomarkers (*p* < 0.05). A majority of participants who had FA concentrations in the ≥95th percentile also exhibited TG, HDL, LDL, and TC levels in the “high” risk classification of developing cardiovascular disease. Future studies profiling individual FA reference ranges in many unique, global cohorts are necessary to develop cut-off values of individual FA concentrations highly related to disease-risk.

## 1. Introduction

Adequate consumption of dietary fatty acids (FA) is essential for the development and maintenance of overall human health and well-being [1]. Due to their wide array of structural and functional capabilities, FA status has been implicated in reducing the risk of a number of chronic diseases [2,3]. There is currently a gap in the field of dietary lipid research as individual FA reference ranges have yet to be established. This fundamental gap hinders the capability of translating reported values from scientific reports for clinical use. Objective measures with established clinical ranges for other lipids such as cholesterol, and triglycerides are commonly used in clinical practices [4]. Developing baseline FA values would provide medical practitioners context regarding the direction and magnitude of change necessary to achieve optimal health in patients with deficient, suboptimal, or even excess amounts of FA. Additionally, FA values which are reported in scientific literature are often inconsistent, using either different units or methods of reporting altogether [5]. In order to address this gap, comprehensive profiles of baseline FA concentrations must be collected and analyzed. Developing validated FA reference ranges while using consistent methods of assessment and reporting are necessary steps in creating generalizable results between individual FA status of different cohorts.

To date, only a small number of studies have comprehensively profiled plasma or serum FA levels in healthy adults with the primary objective of developing reference ranges [6,7,8,9]. No studies to our knowledge have comprehensively examined FA status of individuals in the Singapore-Southeast Asia region of the world. Due to its unique geographical location as an island country located between Malaysia and Indonesia, Singapore has a unique population consisting of three predominant ethnic groups: Chinese, Malays, and Indian [10]. Independent of ethnicity, rates of obesity in Singapore are steadily increasing as their typical diets are very energy-dense [11]. A rise in western-style fast food and high rates of Singaporeans typically going out to eat, has been positively correlated with these dietary changes [11,12]. Additionally, the rise in large-scale production of palm oil, high in the saturated fatty acid (SFA) palmitic acid (PA) and the monounsaturated fatty acid (MUFA) oleic acid (OA), from Indonesia and Malaysia have contributed to high levels of consumption in the region [13]. A number of studies have found significant correlations between FA status and the risk of chronic disease specifically in Singapore Chinese adults [14,15,16,17,18,19,20,21,22,23]. On a global scale, countries in Southeast Asia have reported far higher omega-3 polyunsaturated fatty acids (PUFA) and SFA dietary consumption than North American countries [24]. This study sought to assess and determine the distribution of sixty-seven individual FA concentrations in healthy, adult Singaporean men and women. The unique variance of FA status and lipid biomarkers associated with overall health status and differences between males and females were examined. Given that certain disease-states are known to alter FA status independent of dietary intake [25], using only healthy participants will reduce potential confounding between FA modifications associated with a disease-state or natural fluctuation due to daily living.

## 2. Materials and Methods

### 2.1. Study Subjects

Participants were originally recruited as described in Bi et al. [26]. In brief, 476 healthy male and female adult Singaporeans were recruited from the general public through advertisements placed around the National University of Singapore and the Clinical Nutrition Research Centre website from June 2014 to January 2017. All participants lived in Singapore for at least the past five years and were excluded if they had been diagnosed with any major disease, were pregnant, taking oral contraceptives or, who were breastfeeding. All participants were instructed to not consume alcohol and caffeine-containing drinks for 24 h prior and to fast prior to data collection. All procedures involving human participants were approved by the National Healthcare Group Domain Specific Review Board (NHG DSRB, Reference Number: 2013/00783), Singapore and the University of Guelph (Research Ethics Board Number: 18-10-049).

### 2.2. Anthropometric Measures

Participants reached the laboratory at 8:30 AM after 10 h of an overnight fast, Anthropometric measurements were obtained in a fasted state. Weight (kg) in light clothing without footwear was measured to the nearest 0.1 kg by using an electrical scale and height (cm) was measured using a stadiometer to the nearest millimeter (Seca 763 digital scale, Birmingham, UK). Body mass index (BMI) (kg/m^2^) was calculated using weight divided by the height squared. Waist circumference (WC) was taken at the smallest WC above the umbilicus or navel and below the xiphoid process. DEXA (QDR 4500A, fan-beam densitometer, Hologic, Waltham, MA, USA) was used for the measurement of fat mass (%). To ensure accuracy of the measurement, all metal items were removed from the participants.

### 2.3. Blood Samples

Blood samples were collected as previously described in Bi et al. [27]. In brief, 10 mL of venous blood was collected into Vacutainers (Becton Dickinson Diagnostics, Franklin Lakes, NJ, USA). Blood samples were separated by centrifugation at 1500× *g* for 10 min at 4 °C within 2 h of being drawn and aliquots were stored at −80 °C until analysis. Fasting lipid parameters including total cholesterol, high-density lipoprotein (HDL), and low-density lipoprotein (LDL) cholesterol were measured using the COBAS c311 (Roche)chemistry analyzer. Measurements were done in duplicate and readings were averaged.

### 2.4. Gas Chromatography Protocol

A modified Folch method was used for lipid extraction for serum samples which has been described in-detail previously [28]. In brief, samples were first thawed on ice, where 50 µL of plasma or serum was pipetted into a 15 mL tube. Stock solution of 2:1 chloroform:methanol (*v*:*v*) with a 3.33 µg/mL of internal standard (C19:0) was prepared in which 3 mL of CHCl3: MeOH was added to the tube and capped tightly with a phenolic PTFE liner. After a one minute of vortex, 550 µL of 0.1 M KCl was added to each tube and briefly vortexed again for another 5–10 s. Next, samples were spun at 1460 rpm for 10 min (at 21 °C) to separate phases. The lower chloroform layer was transferred into a clean 15 mL glass tube and dried down under a gentle stream of nitrogen. After, 300 µL of hexane and 1 mL 14% BF3-MeOH was added and followed with a vortex of 5–10 s. Methylation was then completed at 100oC for 60 min in oven, where it was then cooled for 10 min at room temperature. Next, 1 mL of double-distilled water and 1 mL hexane was added to stop the methylation process. After a one-minute vortex, the sample was once again spun down at 1460 rpm for 10 min (at 21 °C) to separate phases. The top hexane layer was then extracted into a clean 2 mL GC vial, dried down under a gentle stream of N2 and then reconstituted in 400 µL of hexane.

Total FA were quantified on an Agilent 7890-A gas chromatography with flame ionization detector and separated on a SP-2560 fused silica capillary column (100 m × 0.25 mm × 0.2 um) (Sigma, Cat # 24056). The splitless inlet (pressure of 19.5 psi and a hydrogen flow of 13.7 mL/min) and detector was set at 250 °C (hydrogen flow of 30 mL/min, air flow of 450 mL/min and nitrogen flow of 10 mL/min), where 1uL of sample was injected onto a splitless inlet set. The oven was initially set at 60 °C, where it was increased to 13 °C/min to 170 °C and was held for 4 min, then followed 6.5 °C/min to 175 °C with no hold to 2.6 °C/min to 185 °C with no hold, to 1.3 °C/min to 190 °C with no hold, to 13.0 °C/min to 240 °C and was held for 13 min. Total run time was 49.78 min per sample.

### 2.5. Fatty Acid Analysis

Peaks were identified and chromatographs were analyzed via OpenLab CDS EZChrom Edition 3.2.1 software. Sixty-seven total FA were included in the pool analyzed and a C19:0 internal standard was used to calculate FA concentrations (μmol/L) and percent composition (%). Non-detectable (nd) values indicate FA below the detectable limit (0.05 µg/mL) of the Agilent 7890-A GC-FID.

### 2.6. Statistical Analysis

Statistical analysis was conducted using SAS University Edition (SAS Institute Inc.; Cary, North Carolina). The range, mean, and percentiles of the concentration of all sixty-seven FA and triglycerides (TG), LDL, HDL, and total cholesterol were profiled. Of the sixty-seven FA analyzed, ten were chosen for further analysis due to their implications to health and prominence in the diet. A Student *t*-test was used to determine statistical differences between male and female participants with results expressed as mean ± standard deviation (SD). A multiple regression model including age, sex, BMI, and FA was conducted to assess the contributions of FA concentrations on TG, LDL, HDL, and total cholesterol. Results were reported as standardized parameter estimates/beta (β) for FA, accounting for the unique variance of the single predictor in the model (FA) while holding the other variables constant. A *p*-value of < 0.05 was considered statistically significant.

## 3. Results

The general characteristics of the entire study population, including differences between males and females, are presented in Table 1. All participants self-reported to be healthy, disease-free, and on average recorded normal BMI values (22.4 kg/m^2^), based on North American and Asian Standards. Significant differences were noted between sexes in measures that are commonly seen in other populations such as: BMI, WC, and fat %. Females had significantly greater levels of HDL cholesterol but both groups had higher concentrations compared to normal values [29]. Both sexes exhibited moderately high concentrations of LDL cholesterol, total cholesterol, and fat %, but TG were well below reported normal levels of 1.7 mmol/L [29].

Table 2 details the mean, range, and percentiles of sixty-seven FA concentrations, expressed as μmol/L for 476 participants. Of this large pool, ten common FA which together can account for more than 90% of the total FA concentration, were selected to be analyzed further due to their prominence in the diet and health implications.

In Table 3, significant differences in the concentrations of SA, PAO, ALA, and DHA (*p* < 0.05) were found between males and females. Females demonstrated higher mean FA concentrations in all four FA which reached significance.

The mean, range, and percentiles of TG, HDL, LDL, and total cholesterol for the entire study population of 476 participants is profiled in Table 4. The cut-off values for lipid biomarkers used by the Singapore Ministry of Health which denotes a health risk are also included [29]. A majority of participants exhibited desirable levels of TG and HDL and borderline/high levels of LDL and total cholesterol (TC).

In Table 5, the variance accounted for by FA concentrations, when holding age, sex and BMI constant, and lipid biomarkers were significant for almost all cholesterol and TG values. Only EPA did not exhibit a significant result with TG (β = 0.00). All ten FA significantly contributed to LDL and total cholesterol. A majority of the FA exhibited significant and positive contributions to HDL cholesterol, but PA, PAO, OA, and ALA did not reach statistical significance. Total FA concentrations all reached significance with circulating triglycerides, HDL, LDL, and total cholesterol.

Based on the cut-off values described in Table 4, the number of participants who exhibited levels of TG, HDL, LDL, or TC which exceeded recommended values are tallied in Table 6. Using the reference ranges of this cohort described in Table 2, FA concentrations in the ≥95th or ≥75th percentile and ≤25th or ≤5th percentile, were evaluated. In the 11 participants with high TG values, a majority of participants were also found to have SFA and MUFA concentrations in the ≥95th percentile, but very few or none had n-3 or n-6 PUFA exceeding the 95th percentile. There were 11 participants with low HDL cholesterol values and a few also had fatty acid concentrations exceeding the 75th percentile or below the 25th percentile. LDL and TC exhibited similar trends, with a large number of participants with FA concentrations in the ≥75th percentile across all ten FA and exceeding LDL and TC cutoffs. Overall, FA concentrations in the ≥95th percentile were far more prevalent in participants exceeding lipid cut-off values than concentrations in the ≤5th percentile.

## 4. Discussion

In this study we determined the percentiles, mean, and ranges of sixty-seven FA collected from 476 healthy, adult Singaporean males and females with ten FA highlighted for further analysis with TG and cholesterol. Significant results between FA concentrations and TG, LDL, HDL, and total cholesterol were seen in almost all of the ten FA highlighted. Mean total FA concentrations reached significance with all four lipid biomarkers but exhibited the weakest relationship with HDL cholesterol. Individual FA concentrations were found to have a significant relationship to all four lipid biomarkers and high concentrations of FA were more often seen in individuals exceeding established cut-off values. Female participants exhibited significantly higher concentrations SA, PAO, ALA, and DHA when compared to males (Table 3), but total FA concentrations did not statistically differ between sexes (Table 1). To our knowledge, the inclusion of sixty-seven FA concentrations is the largest pool of FA to be examined to date for the purposes of profiling FA reference ranges in any cohort.

### 4.1. Individual Fatty Acid Reference Ranges

A very small number of studies have, in detail, profiled FA concentrations in healthy adults for the purposes of developing reference ranges [6,7,8,9]. Glew et al. [30] have also reported mean percent composition of twenty-six FA from serum samples collected in healthy adult males and females from Northern Nigeria. Of these studies, Bradbury et al. [9] and Glew et al. [30] reported their values in percent composition, whereas absolute concentrations were reported in our study. Sergeant et al. [7] (n = 93; obese American women) and Abdelmagid [6] (n = 826; healthy, multiethnic Canadians) also reported their results as absolute concentrations (μmol/L) which allows for the most direct and reliable comparisons to our results. Percent composition is determined by the percentage of one FA from the total pool of FA included in the analysis; a smaller pool analyzed results in a larger apparent percentage of FA [5]. Large fluctuations in a single FA could significantly modify the total value of the pool and subsequent percent composition seen in other FA when no changes in concentration actually occurred. Absolute concentrations of an FA are independent of changes in the pool analyzed and have a larger dynamic range of values, as percent is limited to a 0–100 scale. Reporting results as absolute concentrations has been found to reduce the loss of statistical significance between FA compared to when values were assessed as weight percentages [31]. Furthermore, relative percent can be derived from absolute concentrations, but the same is not possible in reverse as the values of the internal standard used in the study are needed to calculate absolute concentration [5]. As our study included over sixty FA, whereas Bradbury et al. [9] and Glew et al. [30] only included up to twenty-six, it is difficult to accurately compare our reported values even in terms of percent composition. Thus, the use of absolute concentrations is recommended for future studies profiling FA reference ranges to improve comparability and generalizability of results.

Three of the five studies mentioned previously have examined adult North American cohorts and the remaining studies examined cohorts from New Zealand and Nigeria [6,7,8,9,30]. To our understanding, ours is the first study to examine FA concentrations of a large cohort of Southeast Asian, specifically Singaporean, adults in this level of detail. Previous work done in a multiethnic Singaporean cohort found plasma FA composition is a strong, reliable biomarker reflecting dietary consumption in epidemiological studies [32]. The mean FA concentration in our study (11,458 μmol/L) more closely resembled that of Sergeant et al. [7], as Abdelmagid [6] reported a mean concentration which was ~40% lower (6948 μmol/L). Of the ten FA highlighted in Table 3; SA, PAO, AA, and ALA concentrations were noticeably lower in our study whereas EPA and DHA were higher compared to Sergeant et al. [7]. A study conducted in NA by Asselin et al. [33] found a cohort of overweight, older participants (BMI = 25.9; age = 59) reported mean plasma FA concentrations (15,169 μmol/L) similar to that of Sergeant et al. [7] and over double that of the cohort in Abdelmagid et al. [6]. The primary difference that may account for the high concentrations reported in Sergeant et al. [7] is their use of obese middle-aged women (BMI = 33.3; age = 56.9) compared to healthy, young university students (BMI = 22.8; age = 22.6) in Abdelmagid et al. [6]. FA concentrations have been found to strongly correlate with increases in BMI, as obese individuals exhibited modified FA status compared to healthy controls [34,35,36]. Furthermore, as one ages, genetic adaptations occur which can modify FA status, primarily omega-3 PUFA [37]. Taken together, the FA concentrations reported in Sergeant et al. [7] may be more reflective of the average FA status of North Americans due to the high prevalence of obesity in recent years. The use of healthy university-aged students in Abdelmagid et al. [6] could reflect FA concentrations lower than what is expected of the average NA and is representative of a cohort that has reduced risk factors of chronic disease.

### 4.2. Anthropometric Measures

The high total FA concentrations seen in the present study are in contrast to the lower total FA reported in Abdelmagid et al. [6], but the BMI are nearly identical to that of the Singaporean cohort. Our cohort, although reporting a standard healthy BMI, WC, and fasting blood glucose, had mean FA concentrations similar to that of the obese group profiled in Sergeant et al. [7]. Cultural and dietary differences are important modifying factors affecting FA concentrations. First, SFA intake is greater in Singapore than that of NA [24]. The countries surrounding Singapore are the largest producers of palm oil on the planet, which is high in PA, OA, and LA [13]. When examining the mean difference between PA, OA, and LA concentrations between our Singaporean cohort and the Canadian cohort of Abdelmagid et al. [6], those three FA alone account for ~75% of the difference in mean total FA concentrations (calculation not shown). Due to the abundance of palm oil produced in Southeast Asia, dietary intakes have increased which have also been associated with a rise in CVD-related deaths [38]. Palm oil has also been found to strongly correlate with increased LDL cholesterol levels [38], which were reported to be borderline high in our study [6]. Similarly, frequency of omega-3 PUFA consumption via fish is low in NA, but consumption in Southeast Asia are some of the highest in the world [24,39]. Nearly 34.3% of adults in Singapore are overweight or obese and 33.1% are not meeting daily physical activity requirements as of 2014 [40,41]. Measures of body fat % in both males and females in the present study were also quite high compared to other Asian cohorts [42]. In Caucasian men and women, an obese BMI classification of 30 kg/m^2^ corresponds to approximately 25% and 35% body fat [43], equating to roughly the same body fat % reported of both males and females in this study (Table 1). This phenomenon of a low BMI, but high body fat % in Singaporean adults has also been described in previous studies [44]. Widely reported differences between Caucasian populations and Asian populations have resulted in the WHO to make amendments to the current BMI cut-offs for Asian cohorts which was adopted by Singapore in 2005 [45,46]. Current WHO BMI cut points for being considered overweight is 23–27.5 kg/m^2^ and obese is ≥27.5 kg/m^2^ for Asian populations [46]. Although the participants included in our study exhibited healthy levels of TG and HDL cholesterol, and were screened for any chronic diseases, their BMI and body fat % should not be considered to be optimal. Due to this unique paradox found in the Singaporean population, interpretations between FA concentrations and health outcomes related to BMI in other reports should be compared cautiously. Thus, the high FA concentrations seen in our study can be interpreted as a risk factor to chronic disease, similar to that seen in other populations with higher reported body fat % and BMI.

### 4.3. Relationship with Lipid Biomarkers

Unlike FA concentrations, established clinical reference ranges for TG and cholesterol have been established and are routinely utilized in clinical practices to assess health outcomes [4]. By evaluating the relationship between FA concentrations and lipid biomarkers, individual FA can be targeted to determine levels which may be deemed adequate or in excess. Mean LDL and total cholesterol levels in this study were found to be borderline high, which is associated with CVD disease risk (3.35 ± 0.92; 5.27 ± 1.03) [29]. HDL levels were classified as being high (1.68 ± 0.43), which has shown to promote cardioprotective effects [29]. In Table 5, only PA, PAO, OA, and ALA did not reach statistical significance with HDL cholesterol, which has also been seen in previous studies [47,48,49]. EPA and DHA were found to significantly and positively contribute in our model with HDL cholesterol, similar to many previous studies citing strong positive correlations between omega-3 PUFA consumption and the antiatherogenic properties of HDL cholesterol levels [50]. Results of our study support the strong positive correlations between PA and SA to LDL and total cholesterol levels reported previously; a risk factor for developing CHD [51]. Compared to other common vegetable cooking oils low in SFA, palm oil, which is widely consumed in Singapore, has been found to increase LDL cholesterol levels significantly and subsequently increasing the risk of developing type II diabetes [38]. Thus, the values reported in Table 5 for the SFA, PA, and SA, could be disproportionately high based on the dietary habits of Singaporean adults. Unfortunately, dietary data was not available for this study.

The cohort of Singaporeans in the current study had generally very healthy TG levels, with only participants in the 95th percentile and up reaching borderline high values (based on Asian cutoffs) and none of our participants were classified as having very high TG levels (Table 4) [29,52]. Results in Table 5 between PA, SA, PAO, OA, and TG concentrations are similar to the findings reported in previous studies [6,53,54]. High levels of TG have been found to increase the risk of developing CVD, atherosclerosis, and are often seen in obese individuals and those with type-2 diabetes [55,56]. Southeast-Asians on average consume some of the highest levels of omega-3 PUFA via fish in the world, which is reflected by the concentrations of EPA and DHA reported in Table 2 [24]. Greater consumption of omega-3 PUFA is positively correlated with HDL cholesterol, and negatively correlated with TG, resulting in a reduction of developing CVD and CHD [50,57,58]. Results in Table 5, however, indicate omega-3 PUFA, EPA, and DHA, displayed significant positive contributions to TG, LDL, and total cholesterol levels which differs to the negative correlations commonly seen in other studies and as reported by the European Food Safety Authority [53,57]. As we reported larger mean concentrations for nearly the entire FA pool profiled compared to that of a very healthy cohort seen in Abdelmagid et al. [6], high concentrations of health promoting FA and lipid biomarkers may be influenced by increased concentrations of all other FA. We posit the increased omega-3 PUFA concentrations seen are still imparting beneficial health effects despite the positive association to TG and cholesterol. This would provide reasoning for the relatively healthy values of TG and cholesterol seen in Table 1, although mean total FA concentrations were similar to that of an obese cohort [7]. By further establishing strong correlations and determining the unique variance between validated biomarkers and FA concentrations, more precise clinical recommendations can be made to reduce chronic disease risk.

Further work is necessary to establish cut-off values of individual FA concentrations that are deemed to be deficient or in excess. A large body of evidence from multiple cohorts across multiple countries is necessary to establish baseline or “normal” values, and whether region, sex, and/or age specific ranges are necessary [59]. In Table 6, we found that participants with concentrations in the tail ends of FA ranges (≥95th percentile and ≤5th percentile) also exceeded current lipid biomarker cut-offs for TG. However, based on the sample size of our study, the ≥95th percentile and ≤5th percentile contained only 24 participants each, not enough to account for the total participants who exceeded LDL or, total cholesterol cut-offs, thus we expanded the fatty acid cutoff to the ≥75th percentile which includes 119 participants. Of the 92 participants who exceeded the cut-off for LDL cholesterol, ~70% of the participants also had FA concentrations in the ≥75th percentile for PA and SA (Table 6). The exploratory findings in Table 6 and the values seen in Table 5 show that FA concentrations outside the normal range exhibit a large overlap in those with lipid-related health risks. Overall, we more often found a higher proportion of participants who exceeded a lipid biomarker cut-off also having a FA concentration in the ≥95th percentile compared to those in the ≤5th percentile. At this point in time, however, we cannot confidently make any conclusions for specific cut-off values designating adequate/inadequate status of individual FA concentrations. FA reference ranges will need to be further profiled in multiple cohorts to assess whether our results are generalizable to the population at large, or specifically Singaporean adults.

### 4.4. Sex Differences

Along with anthropometric measures, we examined concentrations of circulating FA between sexes, (Table 1 and Table 3). Although Singapore consists of a unique ethnic make-up, where cultural differences can influence dietary habits [10], our population was >90% Singaporean Chinese and thus, ethnic differences were not evaluated. Based on the World Health Organization Global Status Report (40) the BMI and fasting blood glucose of both males and females in this cohort were below national averages, but males were considered to be overweight (23.3 kg/m^2^) based on the BMI cut-offs for Asian populations [46].

We observed mean values for BMI and WC to be below the risk cut-off of an Asian population for developing CVD, diabetes, and dyslipidemia in both sexes, but a number of participants in the higher percentiles shown in Table 4 fell into a risk category [60]. Sex-specific differences in FA status have been shown on a number of occasions, with our results showing women exhibiting significantly greater FA concentrations in Table 3 [35,61,62,63]. In Abdelmagid et al. [6] when males and females were compared PAO, ALA, and DHA were also all significantly greater except, males had greater concentrations of ALA. In terms of percent composition, Glew et al. [30] found no significant difference between males and females in any of the ten FA highlighted in our study. Independent of dietary intake, females have exhibited a greater genetic capacity to endogenously produce omega-3 PUFA [64]. The use of oral contraceptives in some studies have found to further impact PUFA metabolism, but more recent studies found no significant effect [65,66]. Additionally, pregnant and breastfeeding females have obvious metabolic differences and subsequent nutritional needs compared to males which has also been found to modify FA status [67]. Our study excluded female participants who were taking oral contraceptives and who were pregnant or breastfeeding during data collection. Although omega-3 PUFA were significantly greater in females in this study, the mean FA concentration between sexes was not significantly different (see Table 1). As participants were nearly all the same ethnicity and from the same geographical area, dietary habits were likely similar. Genetic differences between sexes may account for some of the variability shown, but dietary consumption still remains the strongest individual modifier of FA concentrations.

### 4.5. Limitations

This study is limited by a lack of dietary data and measures of physical activity recorded. Both physical activity and dietary consumption, particularly of fat, are modifiers of serum FA concentrations. Inclusion of these two variables in the linear regression model would have provided a more sensitive analysis for assessing the unique variance of FA concentrations to lipid biomarkers. Including all ten FA in the linear regression model violated the collinearity tolerance and VIF, which dietary data may have provided a better understanding as to why specific FA were related due to which foods were consumed.

## 5. Conclusions

In conclusion, our study provides valuable knowledge for the growing desire to develop individual FA reference ranges for assessing chronic disease risk. We are one of the few studies to examine a cohort outside of NA which showcases individual FA concentrations. Inclusion of additional cohorts with very high or low concentrations of FA will help establish a greater understanding of FA levels that may be reflective of chronic disease states.

## Figures and Tables

**Table 1 nutrients-13-00435-t001:** General characteristics of study population.

	Total Population	Males	Females	*p*-Value
Population (#)	476	186	290	/
Age (yrs)	38.9 ± 14.6	38.41 ± 14.6	39.21 ± 14.7	0.5601
BMI (kg/m^2^)	22.4 ± 3.4	23.3 ± 2.9	21.8 ± 3.6	<0.0001 *
Waist Circumference (cm)	73.9 ± 9.2	79.3 ± 8.1	70.5 ± 8.2	<0.0001 *
DEXA Fat (%)	31.0 ± 7.7	24.7 ± 5.7	35.1 ± 5.8	<0.0001 *
Glucose (mmol/L)	4.55 ± 0.49	4.63 ± 0.51	4.50 ± 0.46	0.0032 *
HDL Cholesterol (mmol/L)	1.68 ± 0.43	1.51 ± 0.35	1.79 ± 0.44	<0.0001 *
LDL Cholesterol (mmol/L)	3.35 ± 0.92	3.41 ± 0.93	3.32 ± 0.91	0.2771
Total Cholesterol (mmol/L)	5.27 ± 1.03	5.21 ± 0.99	5.31 ± 1.05	0.2868
Triglycerides (mmol/L)	0.97 ± 0.45	1.01 ± 0.44	0.95 ± 0.46	0.1328
Total Serum Fatty Acids (μmol/L)	11,458 ± 2478	11,379 ± 2422	11,509 ± 2515	0.5775

General characteristics of the study population were compared between males and females using a Student *t*-test. * Considered statistically significant at *p* < 0.05. Data represented as mean ± standard deviation. DEXA; Dual-energy X-ray absorptiometry scan.

**Table 2 nutrients-13-00435-t002:** Range, mean, and percentiles of fatty acid concentrations (μmol/L) of serum total lipids.

	Range				Percentile						
Fatty Acid	Min.	Mean	SD	Max.	5	10	25	50	75	90	95
12:0 (Lauric)	t	10.9	12.4	160.0	t	3.5	5.2	7.9	13.2	20.9	29.0
14:0 (Myristic)	25.4	82.8	47.0	342.7	33.6	39.5	48.9	69.8	100.0	146.1	180.1
15:0 (Pentadecanoic)	t	39.2	28.2	207.7	9.6	11.9	16.7	26.4	61.9	77.4	87.8
16:0 (Palmitic)	1400.4	2667.5	625.1	6161.0	1809.8	1991.3	2256.2	2563.0	2967.9	3529.2	3908.3
17:0 (Margaric)	t	37.8	15.4	162.0	17.6	22.2	28.1	35.6	45.0	56.2	66.3
18:0 (Stearic)	415.2	745.2	155.5	1550.6	522.5	568.8	641.5	720.9	828.5	946.0	1021.8
20:0 (Arachidic)	1.8	20.1	5.3	50.3	12.8	14.0	16.6	19.5	22.8	26.4	29.8
21:0	0.7	8.7	7.2	120.1	3.5	4.3	5.7	7.4	9.8	13.0	16.9
22:0 (Behenic)	11.0	45.2	11.6	91.9	28.6	31.1	37.5	44.5	52.0	59.1	65.5
23:0	t	7.0	9.2	36.3	t	t	t	t	15.5	20.7	23.0
24:0 (Lignoceric)	5.0	42.7	11.2	94.1	26.2	29.4	34.9	41.7	50.2	56.0	61.7
12:1c11	nd	nd		nd	nd	nd	nd	nd	nd	nd	nd
14:1c9 (Myristoleic)	t	10.4	9.2	46.6	t	t	2.8	9.1	17.1	22.3	25.9
15:1c10	nd	nd		nd	nd	nd	nd	nd	nd	nd	nd
16:1c9 (Palmitoleic)	6.5	196.0	84.2	526.4	88.8	105.9	135.1	180.2	238.1	309.0	359.4
17:1c10	nd	nd		nd	nd	nd	nd	nd	nd	nd	nd
18:1c9 (Oleic)	263.1	2095.7	633.1	5070.1	1311.1	1431.0	1683.2	1975.6	2389.2	2969.2	3381.2
18:1c11 *(cis-V*accenic)	13.1	163.9	43.2	355.9	106.5	115.7	134.7	159.5	186.7	218.8	240.3
18:1c12	t	4.2	3.3	17.3	t	t	2.1	3.9	5.8	8.1	10.8
18:1c13	t	t		t	t	t	t	t	t	t	t
18:1c14	t	t		t	t	t	t	t	t	t	t
19:1c10	nd	nd		nd	nd	nd	nd	nd	nd	nd	nd
20:1c5	t	6.3	2.9	18.8	3.0	3.3	4.3	5.6	7.4	10.7	12.2
20:1c8	t	3.0	1.6	18.0	1.2	1.5	2.0	2.7	3.7	4.8	5.9
20:1c11 (Gondoic)	2.6	16.2	5.5	52.5	9.8	10.8	12.7	15.4	18.5	23.0	26.5
22:1n9 (Erucic)	t	4.2	4.8	74.3	t	1.9	2.4	3.3	5.2	7.9	8.9
24:1n9 (Nervonic)	2.6	63.9	17.7	187.1	41.1	44.1	52.7	61.6	72.7	84.2	92.3
16:1t9 (Palmitelaidic)	t	7.9	5.7	27.1	2.0	2.2	2.9	6.8	10.9	16.3	19.9
18:1t4	t	0.2	1.4	26.8	t	t	t	t	t	t	1.1
18:1t5	t	6.6	8.8	62.2	t	t	t	3.6	9.1	17.9	25.0
18:1t6-8	t	3.3	2.5	16.5	0.7	0.9	1.5	2.7	4.6	6.5	8.4
18:1t9 (Elaidic)	t	6.7	3.3	47.0	2.7	3.5	4.7	6.3	8.1	10.0	11.9
18:1t10	t	4.4	3.1	29.7	1.1	1.8	2.6	3.6	5.4	7.5	9.4
18:1t11(*trans-V*accenic)	t	7.1	5.0	59.4	1.7	2.4	3.9	6.0	8.9	12.1	15.9
18:1t12	t	4.4	3.8	59.9	1.5	1.9	2.7	3.6	5.1	7.8	9.8
18:1t13	t	11.0	7.0	59.3	4.3	5.1	6.9	9.4	13.2	18.0	21.9
18:1t16	t	3.6	2.4	26.0	t	t	2.4	3.4	4.6	6.4	7.5
18:2tt	t	4.1	4.4	25.9	t	t	0.9	2.7	5.5	10.4	13.6
18:2t9t12 (Linoelaidic)	t	1.2	3.0	25.6	t	t	t	t	t	4.6	6.7
18:2c9t13	t	1.1	2.6	17.8	t	t	t	t	t	4.9	7.0
18:2ct	t	5.2	3.2	24.1	2.0	2.5	3.3	4.4	6.0	8.8	11.7
18:2c9t12	1.0	13.7	4.6	38.5	8.1	9.4	10.8	12.6	15.6	20.0	23.2
18:2t9c12	t	7.9	3.2	21.0	3.2	4.2	5.8	7.7	9.5	11.9	13.4
18:2c9c14	t	1.3	2.4	18.2	t	t	t	t	2.1	3.9	6.3
18:2c9c15 (Mangiferic)	t	2.5	3.6	21.3	t	t	t	t	4.5	7.2	9.2
18:2c9t11 (Rumenic)	t	8.7	2.6	23.4	4.8	6.2	7.4	8.5	9.7	11.5	13.1
18:2c11t13 (CLA)	t	1.0	2.0	10.7	t	t	t	t	t	4.3	5.9
18:2t10c12 (CLA)	t	0.6	1.8	13.0	t	t	t	t	t	2.6	5.1
18:2c/c isomer (CLA)	t	0.3	1.2	12.7	t	t	t	t	t	t	1.7
18:2c/c isomer (CLA)	t	2.1	2.4	16.2	t	t	t	1.7	3.0	4.7	6.6
18:2tt (CLA)	t	8.9	4.4	38.4	0.6	4.7	6.8	8.6	10.7	13.1	15.2
18:2n6 (Linoleic acid)	1652.5	3700.2	777.5	7244.2	2559.2	2766.6	3181.0	3640.6	4090.3	4771.7	5081.2
18:3n6 (γ-linolenic)	2.9	27.0	19.7	124.5	7.5	8.7	12.7	22.5	35.0	51.6	63.2
20:2n6 (Eicosadienoic)	2.5	24.0	7.8	52.7	12.6	15.8	18.8	22.8	28.4	33.6	38.8
20:3n6 (Dihomo-γ-linolenic)	2.0	115.3	46.3	273.9	56.6	64.0	80.1	106.4	141.4	181.1	206.4
20:4n6 (Arachidonic)	104.3	685.8	177.4	1538.7	433.7	480.0	573.0	673.9	784.6	908.3	973.5
22:2n6 (Docosadienoic)	t	7.9	4.8	33.5	2.5	3.2	4.6	6.5	10.3	14.3	16.7
22:4n6 (Adrenic)	3.6	17.7	5.9	50.4	9.3	11.1	14.1	17.2	20.9	25.2	27.4
22:5n6 (Docosapentaenoic)	3.7	19.7	6.2	44.7	11.0	12.6	15.4	19.0	23.4	27.5	31.1
20:3n9 (Mead)	3.0	8.9	4.1	29.4	4.3	4.9	6.2	8.1	10.2	13.9	17.1
18:3n3 *(*α-linolenic)	7.3	57.6	24.7	175.4	28.7	33.0	41.1	52.4	69.7	91.6	103.3
18:4n3 (Stearidonic)	t	t		t	t	t	t	t	t	t	t
20:3n3 (Dihomolinoleic)	t	16.6	11.3	47.1	t	2.3	4.5	19.2	26.2	30.9	33.8
20:5n3 (Eicosapentaenoic)	4.2	93.0	80.9	596.3	20.1	29.0	40.0	68.2	115.2	193.42	285.1
22:3n3 (Docosatrienoic Acid)	nd	nd		nd	nd	nd	nd	nd	nd	nd	nd
22:5n3 (Docosapentaenoic)	14.7	43.5	15.5	118.7	23.7	27/0	32.5	40.7	50.5	65.3	71.5
22:6n3 (Docosahexaenoic)	44.9	255.5	94.4	663.4	121.9	150.7	193.56	240.5	304.0	391.4	430.1

Reference ranges of sixty-seven fatty acids. 19:0 was used as an internal standard and not included in the table. CLA; conjugated linoleic acid t; Trace values recorded. nd; Non-detectable levels.

**Table 3 nutrients-13-00435-t003:** Fatty acids concentrations in males and females.

Fatty Acid	Common Name	Males (n = 186)	Females (n = 290)	*p*-Value
16:0	Palmitic acid (PA)	2683 ± 630.6	2658 ± 622.4	0.66
18:0	Stearic acid (SA)	716 ± 145.2	764 ± 159.2	0.001 *
16:1c9	Palmitoleic acid (PAO)	185 ± 82.5	203 ± 84.7	0.03 *
18:1c9	Oleic acid (OA)	2117 ± 645.0	2082 ± 625.9	0.55
18:2n6	Linoleic acid (LA)	3652 ± 737.4	3731 ± 801.9	0.28
20:4n6	Arachidonic acid (ARA)	687 ± 176.2	685 ± 178.5	0.86
18:3n3	Alpha-linolenic acid (ALA)	53.9 ± 23.0	60.0 ± 25.4	0.008 *
20:5n3	Eicosapentaenoic acid (EPA)	84.4 ± 74.2	98.5 ± 84.7	0.06
22:5n3	Docosapentaenoic acid (DPA)	42.6 ± 14.8	44.1 ± 0.9	0.3
22:6n3	Docosahexaenoic acid (DHA)	240 ± 86.8	266 ± 97.8	0.003 *

Fatty acid concentrations of ten biologically relevant fatty acids are compared between male and female participants by the Student’s *t*-test. Results are presented as mean ± standard deviation. * Considered statistically significant at *p* < 0.05.

**Table 4 nutrients-13-00435-t004:** Reference ranges and clinical cut-offs of lipid biomarkers.

	Range			Percentile						
Class	Min	Mean ± SD	Max	5	10	25	50	75	90	95
TG	0.36	0.97 ± 0.45	3.06	0.50	0.55	0.66	0.87	1.13	1.64	1.91
HDL	0.80	1.68 ± 0.43	3.44	1.07	1.16	1.38	1.65	1.91	2.29	2.48
LDL	0.99	3.35 ± 0.92	7.64	2.06	2.29	2.74	3.23	3.87	4.48	4.99
TC	2.70	5.27 ± 1.03	9.44	3.74	4.08	4.58	5.17	5.85	6.67	7.05
Class	Desirable ^a^		Borderline/High				Very High			
TG	1.7–2.2 mmol/L		2.3–4.4 mmol/L				≥4.5 mmol/L			
HDL	1.0–1.5 mmol/L		>1.6 mmol/L							
LDL	2.6–3.3 mmol/L		3.4–4.8 mmol/L				≥4.9 mmol/L			
TC	<5.2 mmol/L		5.2–6.1 mmol/L				≥6.2 mmol/L			

The top half of the table displays the range of values for the four lipid biomarkers evaluated for all participants in this cohort. The bottom half of the table details the clinical cut-off values associated with chronic health risk of each of the four lipid biomarkers as classified by Ministry of Health for Singapore. ^a^ Tai et al. [29]. HDL; high-density lipoprotein, LDL; low-density lipoprotein, TC; total cholesterol, TG; triglycerides.

**Table 5 nutrients-13-00435-t005:** Multiple regression model assessing the unique variance between select fatty acids and lipid biomarkers.

Fatty Acid	Common Name	Triglycerides	HDL	LDL	Total Cholesterol
		β	β	β	β
16:0	Palmitic acid (PA)	0.78 *	0.02	0.55 *	0.64 *
18:0	Stearic acid (SA)	0.64 *	0.18 *	0.60 *	0.72 *
16:1c9	Palmitoleic acid (PAO)	0.70 *	−0.05	0.31 *	0.38 *
18:1c9	Oleic acid (OA)	0.81 *	−0.10	0.46 *	0.50 *
18:2n6	Linoleic acid (LA)	0.45 *	0.24 *	0.62 *	0.72 *
20:4n6	Arachidonic acid (ARA)	0.17 *	0.25 *	0.43 *	0.37 *
18:3n3	Alpha-linolenic acid (ALA)	0.61 *	−0.06	0.29 *	0.34 *
20:5n3	Eicosapentaenoic acid (EPA)	0	0.20 *	0.22 *	0.29 *
22:5n3	Docosapentaenoic acid (DPA)	0.33 *	0.19 *	0.40 *	0.49 *
22:6n3	Docosahexaenoic acid (DHA)	0.27 *	0.21 *	0.44 *	0.52 *
	TOTAL	0.67 *	0.12 **	0.59 *	0.69 *

A multiple regression model was conducted to examine the relationship between ten biologically relevant fatty acids and the four lipid biomarkers measured. The model included the fatty acid along with age, sex, and BMI. The β reported is for the relationship between the fatty acid of interest and the lipid biomarker. * *p* < 0.001, ** *p* < 0.05. HDL; high-density lipoprotein, LDL; low-density lipoprotein.

**Table 6 nutrients-13-00435-t006:** Exploratory investigation of the upper and lower percentile fatty acid concentrations in participants who exceed risk cut-offs for lipid biomarkers.

Lipid Class	Exceeded Cut-Off ^a^ (#)	Percentile	16:0	18:0	16:1c9	18:1c9	18:2n6	20:4n6	18:3n3	20:5n3	22:5n3	22:6n3
TG	11	≥95% ^b^	10	8	7	11	3	0	6	0	1	1
		≤5% ^b^	0	0	0	0	0	0	0	0	0	0
HDL-Low	11	≥75% ^c^	4	3	2	4	1	0	3	2	2	2
		≤25% ^c^	3	5	3	1	4	5	4	4	6	4
LDL	91	≥75% ^c^	62	63	38	54	65	46	44	39	52	52
		≤25% ^c^	1	1	6	0	1	9	5	9	5	5
TC	82	≥75% ^c^	54	60	37	50	66	52	43	43	53	52
		≤25% ^c^	0	0	5	1	0	7	4	7	2	4

Values represent number of participants who exceed the risk cut-off values for lipid biomarkers and are also in the upper/lower 5th and 25th percentile of fatty acids. These percentiles were chosen to best capture the totality of participants who exceeded biomarker cut-off values. ^a^ TG ≥ 2.3mmol/L; HDL-Low < 1.0 mmol/L; LDL ≥ 4.1 mmol/L; TC ≥ 6.2 mmol/L. ^b^ Each fatty acid has 24 total participants in each of the upper and lower 5th percentile. ^c^ Each fatty acid has 119 total participants in each of the upper and lower 25th percentile, except 18:3n3 which has 120. HDL; high-density lipoprotein, LDL; low-density lipoprotein, TC; total cholesterol, TG; triglycerides.

## Data Availability

The data presented in this study is available on request from the corresponding author on reasonable request. The data is not publicly available due to privacy or ethical restrictions.

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
