# Peer review of "Development of Fatty Acid Reference Ranges and Relationship with Lipid Biomarkers in Middle-Aged Healthy Singaporean Men and Women"

_nutrients, 2021, doi:10.3390/nu13020435_

Round 1

Reviewer 1 Report

Review of

Development of Fatty Acid Reference Ranges and Correlations to Lipid Biomarkers in Healthy Singaporean Men and Women.

The authors have carefully collected extensive data on fatty acid profiles in a Southeast Asian population of almost 500 participants and correlated these to plasma lipid biomarkers, namely, HDL, LDL, triglycerides and total cholesterol. The data will certainly be very useful in establishing reference ranges for specific populations. It is laudable that both men and women have been included.

Specific comments and questions:

  • Under 2.1 it is said that participants had to restrict alcohol and caffeine 24h prior to sampling and had to fast prior to data collection. It is not clear, what restrict means here, and the duration of the fast is also not given. Please state clearly, what the restriction for alcohol and caffeine were (none at all 24 hours prior to sampling or limited intake only). Also, state how long participants were expected to fast. When were samples taken? Always in the morning or another fixed part of the day, or did sampling take place throughout the whole day.
  • Also, in 2.1. it is said that pregnant women were excluded, but later in the discussion, it is said, that women on oral contraceptives or who were breastfeeding, were excluded, too. This should be clearly stated in 2.1 already.
  • Under 2.6, the statistical analyses are explained. I am not a statistician, and it is not clear to me what beta-weight (b)of age exactly stands for. It is also shown in Table 5. I assume that it tells me that certain correlation become stronger or weaker with increasing age, but it would be useful to have this explained in more detail.
  • BMI – interpretation. Table 1 provides the general characteristics of the study population. In the accompanying text, it is stated that all participants were within the normal range. This is true, when assuming North-American or Caucasian BMIs, but for Asian people, a BMI above 22.9 is already associated with a higher risk for diabetes. Later, in the discussion, the authors discuss this extensively, but in my opinion, this distinction should already be introduced here, directly after Table 1. Since in the discussion, some of the findings are also interpreted in the light of overweight of participants, it might be more useful to separate the participants into three or four groups based on Asian recommendations for BMI (below normal, normal, overweight, obese) than give the percentiles in Tables 2 and 4, that I do not find very helpful.
  • In the text between Tables 1 and 2, it is said that females and males had higher HDL concentrations compared to normal values – but no references are provided. To which normal values do you refer.
  • Table 2: I assume that in the definitive article, the layout of this table will be improved, e.g. by using a smaller font size? In the current format it is very difficult to read. The definition for t =trace amounts is not very clear. In the Materials section t is said to mean 0 or below detection limit of the device. Below the table it says: trace amounts detected. What is it? Trace amounts – in that case give an upper limit, or below detection level? Please clarify.
  • In the text between Tables 2 and 3, it says: Of this large pool, ten common FA which consisted of more than 90% of total FA… I assume that the authors meant to say: ten common FA which together accounted for more than…
  • Please introduce all abbreviations (e.g. SA,PAO, ALA could be introduced in Table 3).
  • Delete the sentence: , with males demonstrating slightly higher means in only PA, OA and ARA. They have no predictive value.
  • The discussion is very long, and often results of the study are repeated. If it stated in the beginning that the study population is normal to overweight, then the comparison to the North-American study with lean, young participants can be simplified. In my opinion, the discussion can easily be shortened by one page.

Reviewer 2 Report

This study sought to assess and determine the distribution of sixty-seven individual FA concentrations in healthy, adult Singaporean men and women. In addition, authors tried to find correlations between FA status and lipid biomarkers associated with overall health status and differences between males and females. Authors provided valuable knowledge for the growing desire to develop individual FA reference ranges for assessing chronic disease risk. However, additional data and manuscript revisions are necessary to improve manuscript.

  1. Authors described that the range, mean, and percentiles of the concentration of all sixty-seven FA, whereas Bradbury et al. [9] and Glew et al. [30] only included up to twenty-six FA. Please describe why 67 FA references are necessary and discuss about the different meanings of other studies. Reviewer thinks that it is necessary to minimize the differences to get reference range of FA.
  2. Bradbury et al. [9] analyzed FA according to age category, ethnicity and BMI category. Please analyze FA according to age category, ethnicity and BMI category.
  3. The unit of FA is mol%.in other study. Please explain the reason using μmol/L and discuss about the unit of FA.
  4. It is necessary to decide FA reference range for assessing chronic disease risk with this study. The mean FA concentration in this study (11,458 μmol/L) more closely resembled that of Sergeant et al. [7], as Abdelmagid [6] reported a mean concentration which was ~40% lower (6,948 μmol/L) μmol/L. Is it important total FA concentration? Are sixty-seven FA or 10 FA in table 3 important for assessing chronic disease risk?

Minor comments

Please change “individual FA” to “FA”.

Please add “middle aged” in the title.

Round 2

Reviewer 2 Report

Authors adequately answered to every question and minor comments.